# Thoughts Are All Over the Place: On the Underthinking of o1-Like LLMs

## Abstract

Large language models (LLMs) such as OpenAI's o1 have demonstrated remarkable abilities in complex reasoning tasks by scaling test-time compute and exhibiting human-like deep thinking. However, we identify a phenomenon we term **underthinking**, where o1-like LLMs frequently switch between different reasoning thoughts without sufficiently exploring promising paths to reach a correct solution. This behavior leads to inadequate depth of reasoning and decreased performance, particularly on challenging mathematical problems. To systematically analyze this issue, we conduct experiments on three challenging test sets and two representative open-source o1-like models, revealing that frequent thought switching correlates with incorrect responses. We introduce a novel metric to quantify underthinking by measuring token efficiency in incorrect answers. To address underthinking, we propose a decoding strategy with thought switching penalty (TIP) that discourages premature transitions between thoughts, encouraging deeper exploration of each reasoning path. Experimental results demonstrate that our approach improves accuracy across challenging datasets without requiring model fine-tuning. Our findings contribute to understanding reasoning inefficiencies in o1-like LLMs and offer a practical solution to align their problem-solving capabilities to human-like deep thinking.

## 1. Introduction

Large Language Models (LLMs), such as OpenAI's o1 (OpenAI, 2024), have revolutionized artificial intelligence by enabling models to tackle increasingly complex tasks. The o1 model and its replicas (Qwen, 2024; DeepSeek, 2025; Kimi, 2025), known for their deep reasoning capabilities,

[1]Anonymous Institution, Anonymous City, Anonymous Region, Anonymous Country. Correspondence to: Anonymous Author <anon.email@domain.com>.

Preliminary work. Under review by the International Conference on Machine Learning (ICML). Do not distribute.

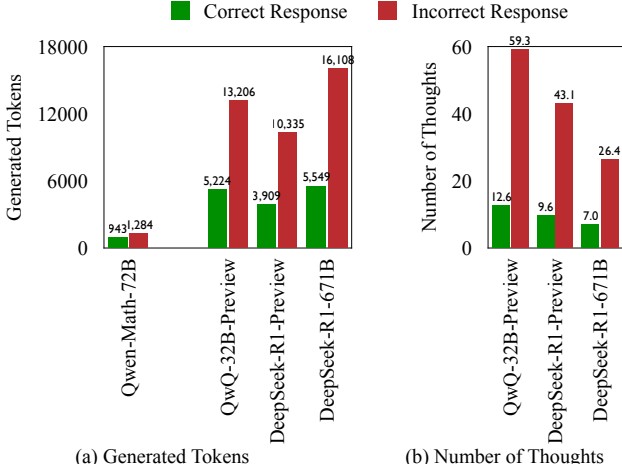

*Figure 1.* Illustration of the **underthinking issue** on the challenging AIME2024 testset: Incorrect responses from o1-like models (e.g., QwQ-32B-Preview and DeepSeek-R1-671B) often involve a higher frequency of switching reasoning strategies compared to correct answers (Figure b). This results in significantly longer response lengths without corresponding improvements in accuracy (Figure a). In contrast, conventional LLMs (e.g., Qwen-Math-72B) do not show significant differences in response length between incorrect and correct answers.

exemplify the potential of LLMs to exhibit human-like deep thinking by scaling test-time computation during problem-solving. These models aim to explore diverse reasoning strategies, reflect on their decisions, and iteratively refine solutions, closely mimicking human cognitive processes.

Despite their successes, a critical yet underexplored question remains: **Are o1-like LLMs thinking deeply enough?** This study provides an initial exploration of this problem. In this work, we investigate a phenomenon we term **underthinking**, which refers to the tendency of o1-like LLMs to prematurely abandon promising lines of reasoning, leading to inadequate depth of thought. To systematically analyze underthinking, we conduct experiments on three challenging test sets (e.g., MATH500, GPQA Diamond, and AIME2024) and two open-source o1-like models with visible long chains of thought (e.g., QwQ-32B-Preview and DeepSeek-R1-671B). Through extensive analyses, we found that underthinking manifests in the following patterns: (1)

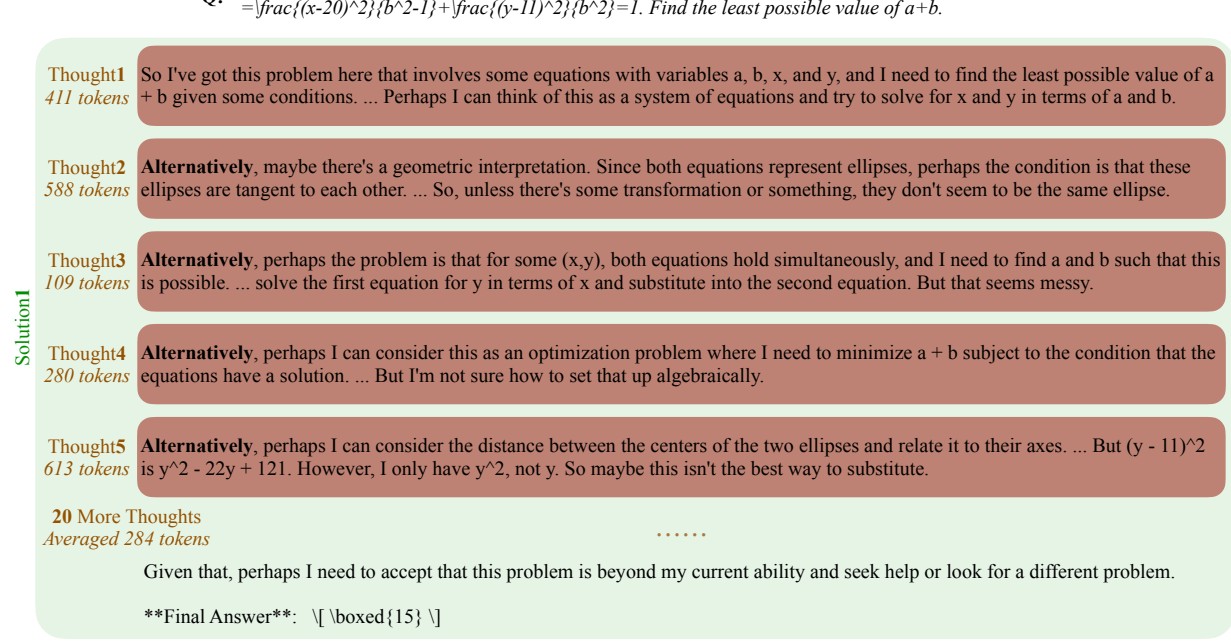

**Q:** *Let a, b, x, and y be real numbers with a>4 and b>1 such that \frac{x^2}{a^2}+\frac{y^2}{a^2-16} =\frac{(x-20)^2}{b^2-1}+\frac{(y-11)^2}{b^2}=1. Find the least possible value of a+b.*

**Solution1**

**Thought1** *411 tokens* — So I've got this problem here that involves some equations with variables a, b, x, and y, and I need to find the least possible value of a + b given some conditions. ... Perhaps I can think of this as a system of equations and try to solve for x and y in terms of a and b.

**Thought2** *588 tokens* — **Alternatively**, maybe there's a geometric interpretation. Since both equations represent ellipses, perhaps the condition is that these ellipses are tangent to each other. ... So, unless there's some transformation or something, they don't seem to be the same ellipse.

**Thought3** *109 tokens* — **Alternatively**, perhaps the problem is that for some (x,y), both equations hold simultaneously, and I need to find a and b such that this is possible. ... solve the first equation for y in terms of x and substitute into the second equation. But that seems messy.

**Thought4** *280 tokens* — **Alternatively**, perhaps I can consider this as an optimization problem where I need to minimize a + b subject to the condition that the equations have a solution. ... But I'm not sure how to set that up algebraically.

**Thought5** *613 tokens* — **Alternatively**, perhaps I can consider the distance between the centers of the two ellipses and relate it to their axes. ... But (y - 11)^2 is y^2 - 22y + 121. However, I only have y^2, not y. So maybe this isn't the best way to substitute.

**20** More Thoughts
*Averaged 284 tokens*                    ......

Given that, perhaps I need to accept that this problem is beyond my current ability and seek help or look for a different problem.

**Final Answer**:  \[ \boxed{15} \]

*Figure 2.* An example of underthinking issue for QwQ-32B-Preview model's output response that consists of 25 reasoning thoughts within a single solution. Generally, thought does not reach a conclusion.

it occurs more frequently on harder problems, (2) it leads to frequent switching between different thoughts without reaching a conclusion in each, and (3) it correlates with incorrect responses due to insufficient exploration of reasoning paths. For example, Figure 1 compares the token usage and number of thoughts of correct and incorrect responses. On average, o1-like LLMs consume 225% more tokens in incorrect responses than in correct ones due to 418% more frequent thought-switching behaviors.

To quantify this phenomenon, we introduce a novel *underthinking metric* that measures token efficiency in incorrect responses by evaluating the proportion of the response that contributes to reaching correct thoughts. Combining the widely-used accuracy metric with the proposed underthinking metric provides a more comprehensive assessment of o1-like models: accuracy measures how often the model can produce *correct responses*, while the underthinking metric evaluates the token efficiency within *incorrect responses* that contributes to reaching correct thoughts.

In response to these findings, we propose a decoding strategy with thought switching penalty (TIP) that discourages premature transitions between thoughts during the generation process. By adjusting decoding penalties for tokens associated with thought switching, the model is encouraged to thoroughly develop each line of reasoning before considering alternatives. Experimental results show that employing TIP improves accuracy across challenging test

sets without requiring additional model fine-tuning.

Our study makes the following contributions:

1. We formally define and characterize the underthinking issue in o1-like LLMs, where models frequently abandon promising reasoning paths prematurely, leading to inadequate depth of reasoning on challenging problems.

2. We introduce a novel metric to evaluate underthinking by measuring token efficiency in incorrect responses, providing a quantitative framework to assess reasoning inefficiencies.

3. We propose a decoding approach with thought switching penalty (TIP) that encourages models to deeply explore each reasoning thought before switching, improving accuracy without additional model fine-tuning.

## 2. Observing Underthinking Issues

In this section, we present a comprehensive analysis of outputs from o1-like models on *challenging math problems*. We begin by illustrating the frequent thinking switch phenomenon observed in responses to these problems, as shown in Figure 2, highlighting how this behavior differs significantly between correct and incorrect answers (Section 2.1). We then show that this phenomenon leads to an inadequate depth of reasoning, causing models to *abandon promising*

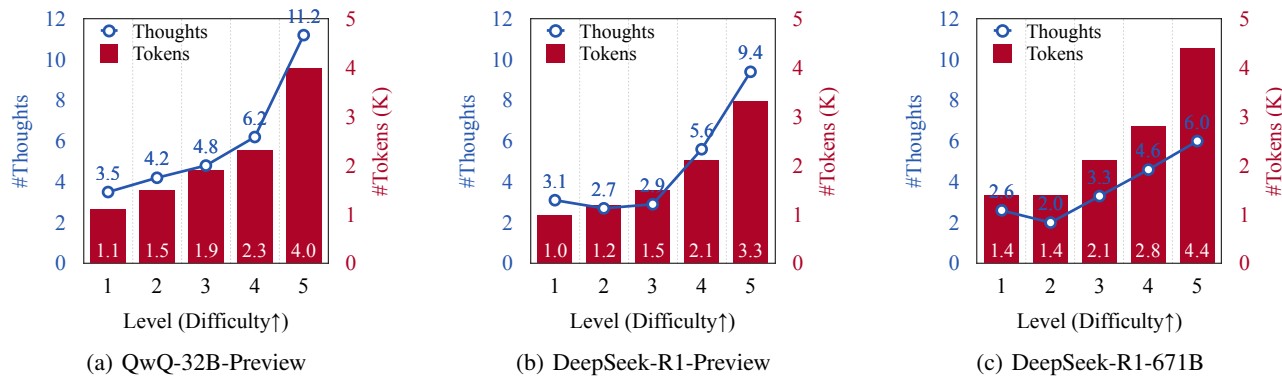

*Figure 3.* Average number of thoughts and tokens in generated responses across different difficulty levels of the MATH500 test set.

*reasoning paths prematurely* (Section 2.2). Based on this observation, we propose a metric to empirically assess the underthinking issues and present empirical results in Section 2.3. We conclude that *o1-like LLMs often underthink when they fail to tackle challenging math problems.*

### 2.1. Frequent Thinking Switch of o1-Like LLMs

We conduct experiments on three testsets:

- **MATH500** (Hendrycks et al., 2021): a challenging dataset consisting of problems from **high school math competitions** across seven subjects (e.g., Prealgebra, Algebra, Number Theory) and difficulty levels based on AoPS (ranging from 1 to 5). Problems in these competitions range from level 1, the easiest, often found in AMC 8 exams, to level 5, like those in AIME.

- **GPQA** (Rein et al., 2023): a **graduate-level** dataset consisting of multiple-choice questions in subdomains of physics, chemistry, and biology. For our experiment, we select the highest quality subset, known as GPQA Diamond (composed of 198 questions).

- **AIME** (MAA Committees): a dataset from the American Invitational **Mathematics Examination**, which tests math problem solving across multiple areas (e.g. algebra, counting, geometry, number theory, and probability). Because AIME 2024 contains only 30 examples, we also considered 60 more examples from AIME 2022 and 2023.

We mainly investigate two widely recognized open-source o1-like models featuring visible long CoT: QwQ-32B-Preview and DeepSeek-R1-671B. We also include DeepSeek-R1-Preview to show the development of R1 series models. Given DeepSeek-R1-Preview's daily message limit of 50 via web interface, we evaluated this model solely on the MATH500 and AIME test sets.

**Definition of Reasoning Thoughts** In this paper, we define *thoughts* as the intermediate cognitive steps within a reasoning strategy produced by the model. O1-like LLMs often switch reasoning thoughts using terms like "alternatively". For instance, as shown in Figure 2, the problem-solving process involves multiple reasoning thoughts, shifting from algebraic manipulation to geometric interpretation and optimization strategies. The ability to switch between different reasoning strategies allows for a broader exploration of potential solutions and demonstrates the flexibility of the model in tackling complex problems. In this study, we provide a comprehensive analysis of the side effects associated with this ability to switch reasoning thoughts.

We utilize the Llama-3.3-70B model to automatically segment a response into reasoning thoughts due to its superior capabilities in both instruction following and mathematical reasoning. Initially, we manually analyzed responses from the QwQ-32B-Preview model to gather expressions indicative of shifts in thought. We then tasked the Llama-3.3-70B model with scanning the entire response to identify all occurrences of such expressions. Furthermore, we asked the model to determine whether these expressions truly signify a change in thought or merely reflect a stylistic pattern in the response. Only the expressions indicating a genuine thought shift were used as separators for reasoning processes.

**o1-Like LLMs Switch Thinking More Frequently on Harder Problems** Figure 3 shows the averaged thoughts and tokens in generated responses across various difficulty levels in the MATH500 test set. Clearly, all models generate more reasoning thoughts with the increase of difficulty level, which is consistent with the growth of generated tokens. This observation suggests that as the complexity of the problems increases, the models tend to switch thoughts more frequently. This behavior implies that o1-like LLMs are able to dynamically adjust their reasoning processes to tackle more challenging problems. The following experiments focus on Level 5 in the MATH500 test set (MATH500-Hard).

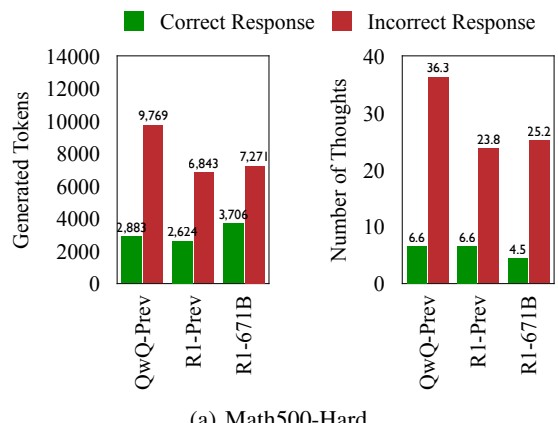

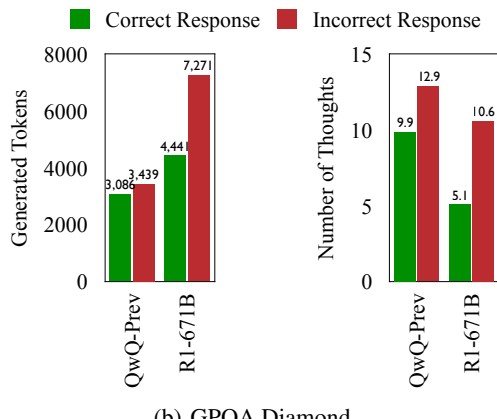

(a) Math500-Hard           (b) GPQA Diamond

*Figure 4.* O1-like LLMs switch thinking more frequently on incorrect responses, thus expend more tokens without contributing to accuracy.

**Increased Thought Switching in o1-Like LLMs during Incorrect Responses** When examining the behavior of o1-like LLMs, we observe a distinct pattern in how they handle incorrect responses. As depicted in Figures 1 and 4, these models exhibit a significant increase in the frequency of thought switching while generating incorrect answers across all test sets. This trend suggests that although the models are designed to dynamically adjust their cognitive processes to solve problems, more frequent thought switching does not necessarily lead to higher accuracy. Essentially, the models may be expending additional computational resources – evidenced by an increase in generated tokens – without achieving more accurate solutions. These insights are crucial because they highlight the need not only to explore additional cognitive pathways when faced with challenges but also to operate in **a more targeted and efficient manner**, thereby improving accuracy even when complex reasoning is required. In the following sections, we empirically validate the inefficiencies associated with frequent thought switching in incorrect responses.

### 2.2. Existence of Underthinking

The behavior of frequent thinking switch in incorrect responses could stem either from (1) genuine underthinking, where the model succeeds in finding promising strategies but fails to stick with them, or from (2) a lack of understanding, prompting it to explore diverse but ineffective approaches. To disentangle these possibilities, we propose an assessment framework that evaluates whether an abandoned reasoning path is actually sufficient to derive a correct answer. By focusing on whether the model can persistently follow and deepen a single, promising line of thought, we can identify instances of underthinking.

**Assessing Thought Correctness** In the example presented in Figure 2, we observe that some early thoughts may lead to the correct answer. For instance, Thought 1 ini-

tiates a correct interpretation by recognizing that the given equations resemble those of ellipses centered at (0,0) and (20,11). Setting the two expressions equal is a valid approach to finding common points $(x, y)$ that satisfy both equations. Instead of concentrating on thoroughly exploring the plausible thought with further algebraic manipulation and optimization techniques, the model frequently shifts its focus and uses approximately 7,270 additional tokens without arriving at a correct answer. Ultimately, it concludes with a guessed answer that lacks support from the extended COT process.

We leverage LLMs to assess whether each thought leads to a correct answer using the following prompt:

---

Problem P = {problem}
Solution Draft S = {split solutions}
Correct Answer A = {expected answer}

1. *Please analyze the relevance between the solution S and the problem P, and conduct some verifications to check the correctness of the solution itself. Please think step by step to give an explanation **EXPLANATION**.*
2. *If you think the solution draft S can lead to the correct answer A of the problem P, please stick to the line of thinking without deviation and carry it through to completion. If you think it cannot yield the correct answer or you're not sure, don't force yourself to give an answer and generate **None**.*
3. *Please tell me honestly how confident you are that you can solve the problem P correctly based on the the solution draft S. Out of 2, please generate your confidence score **CONFIDENT_SCORE**.*

Please output **EXPLANATION** and **CONFIDENT_SCORE** according to the following format:
EXPLANATION: \boxed{}
CONFIDENT_SCORE: \boxed{}

---

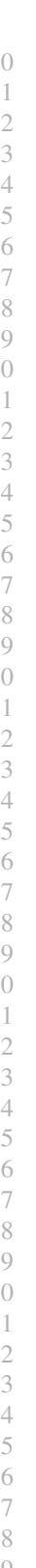
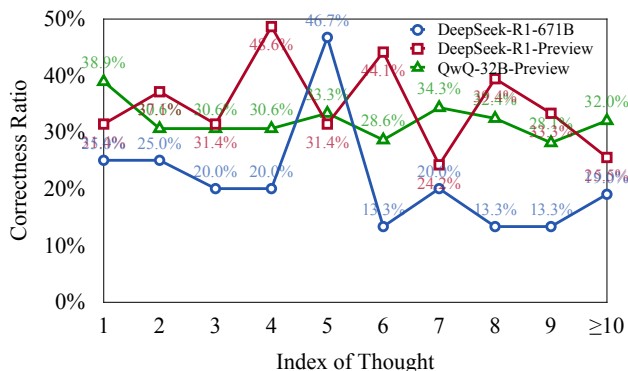
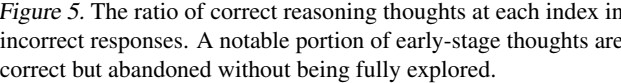

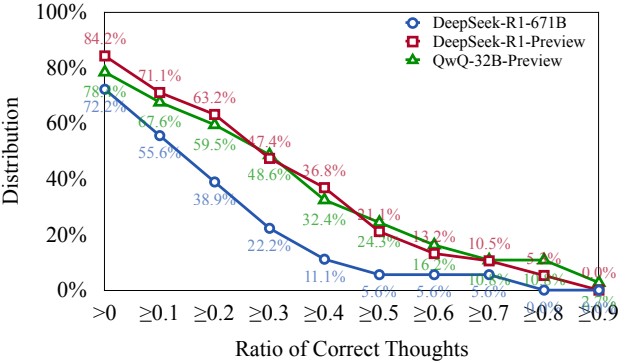

*Figure 5.* The ratio of correct reasoning thoughts at each index in incorrect responses. A notable portion of early-stage thoughts are correct but abandoned without being fully explored.

*Figure 6.* The distribution of thought correctness ratio in incorrect responses. Most incorrect responses contain correct thoughts.

Specifically, we use two models distilled from DeepSeek-R1-671B based on `Llama` and `Qwen` – *DeepSeek-R1-Distill-Llama-70B* and *DeepSeek-R1-Distill-Qwen-32B*, which achieve new state-of-the-art results for dense models across various reasoning benchmarks. If at least one model generates a confidence score of 2 for a thought, we regard it as a correct thought.

We evaluate the accuracy of our assessment approach using responses generated by Qwen-32B-Preview for 90 instances from the AIME 2022, 2023, and 2024 test sets. We utilize the final thought in each response as the test example and its correctness as the ground-truth label. To ensure a fair comparison, we randomly streamline correct thoughts to match the average length of incorrect thoughts. Ultimately, we have 35 correct thoughts with an average length of 278.1 tokens and 55 incorrect thoughts with an average length of 278.3 tokens. Our assessment approach achieves accuracies of 82.9% for correct examples and 81.8% for incorrect examples, demonstrating its effectiveness.

**Early-Stage Thoughts Are Correct but Abandoned in Incorrect Responses** Figure 5 depicts the ratio of correct thoughts at each index in incorrect responses on the three challenging test sets. The analysis highlights a critical insight into the phenomenon of underthinking. Specifically, a notable proportion of initial thoughts across various models were correct but were not pursued to completion. This tendency to abruptly shift away from these promising thoughts indicates an inadequate depth of reasoning, where potentially correct solutions are prematurely abandoned before being thoroughly explored. This observation suggests a need for enhancing the models' ability to persistently explore a specific line of reasoning deeply and accurately before opting to switch to alternative thought processes.

**Most Incorrect Responses Contain Correct Thoughts**
Figure 6 illustrates the distribution of thought correctness ratios in incorrect responses from various models. We observe that over 70% of incorrect responses contain at least one correct thought. Furthermore, in more than 50% of these responses, over 10% of the thoughts are correct. Combined with observations from Figure 5, this suggests that while o1-like models can initiate correct reasoning pathways, they may struggle to continue these pathways to reach the correct conclusion. This highlights the importance of encouraging models to maintain and expand their **initial correct thoughts** to synthesize them into accurate final answers. These insights lead us to propose an underthinking metric based on the presence of the first correct thought in the subsequent section.

### 2.3. Empirical Underthinking Results

In this section, we propose a metric for empirically assessing underthinking issues based on token efficiency, complementing the widely used accuracy metric.

**Underthinking Metric** Intuitively, if a model generates a correct thought at an early stage and then switches to other thoughts without reaching a correct answer, the tokens generated thereafter do not contribute to reaching a correct solution and are considered inefficient due to underthinking. The underthinking score, denoted as $\xi_{UT}$, is defined as:

$$\xi_{UT} = \frac{1}{N} \sum_{i=1}^{N} \left( 1 - \frac{\hat{T}_i}{T_i} \right) \quad (1)$$

Here, $N$ represents the number of instances in a given test set where the evaluated model generates **incorrect responses**. $T_i$ is the total number of tokens in the $i$-th incorrect response, and $\hat{T}_i$ is the number of tokens from the beginning of that response up to and including the first correct thought. If there is no correct thought in the $i$-th response,

*Table 1.* Underthinking scores on challenging testsets.

| Models | Accuracy($\uparrow$) | UT Score ($\downarrow$) |
|---|---|---|
| **MATH500-Hard (Level 5)** | | |
| QwQ-32B-Preview | 84.3 | 58.2 |
| DeepSeek-R1-Preview | 83.6 | 61.5 |
| DeepSeek-R1-671B | 92.5 | 65.4 |
| **GPQA Diamond** | | |
| QwQ-32B-Preview | 59.6 | 48.3 |
| DeepSeek-R1-671B | 73.2 | 58.8 |
| **AIME2024** | | |
| QwQ-32B-Preview | 46.7 | 65.0 |
| DeepSeek-R1-Preview | 46.7 | 75.7 |
| DeepSeek-R1-671B | 73.3 | 37.0 |

$\hat{T}_i = T_i$, indicating that the model lacks an understanding of this problem, leading it to explore diverse but ineffective approaches. Therefore, it cannot be considered underthinking. Consider Figure 2 as an example: the first reasoning thought can reach a correct answer if fully explored, with $\hat{T} = 411$. Consequently, $\xi_{UT} = 1 - \frac{411}{7681} = 0.946$, which can be considered extremely inefficient, reflecting a high underthinking score.

The metric $\xi_{UT}$ quantifies the extent of underthinking by measuring the token efficiency in generating effective content within an incorrect response. Specifically:

- A lower value of $\xi_{UT}$ indicates higher token efficiency, meaning that a greater proportion of tokens in incorrect responses contribute towards reaching a correct thought before switching to another thought. This suggests that the model is more efficient in its token utilization even when it fails to provide a correct answer.

- Conversely, a higher value of $\xi_{UT}$ signifies lower token efficiency, indicating that a larger proportion of tokens do not contribute effectively towards generating a correct thought. This reflects greater underthinking, where the model may generate redundant or irrelevant tokens by frequently switching thoughts.

**Empirical Results**  Table 1 provides insights into model performance across challenging test sets, evaluating both accuracy and underthinking (UT) scores. Clearly, all o1-like LLMs suffer from significant underthinking issues, although there are considerable differences across models and test sets. The results reveals that the relationship between model accuracy and underthinking varies across different datasets. On the MATH500-Hard and GPQA Diamond datasets, higher accuracy achieved by the superior DeepSeek-R1-671B model is accompanied by higher

UT Scores, indicating more underthinking in incorrect responses. This suggests that while the model is more capable overall, it may produce longer but less effective reasoning when uncertain, possibly due to exploring multiple incorrect reasoning paths without efficiently converging on the correct solution. Conversely, on the AIME2024 test set, the DeepSeek-R1-671B model not only attains higher accuracy but also exhibits a lower UT score, reflecting less underthinking and greater token efficiency. This implies that the model's reasoning remains focused and effective even when it does not arrive at the correct answer, perhaps due to better alignment with the problem types and reasoning processes required by the AIME2024 task.

These findings illustrate that underthinking behavior is sensitive to the nature of the dataset and the tasks involved. The larger model's superior capabilities do not uniformly translate to less underthinking across all tasks. In some cases, increased model capacity leads to more elaborate but inefficient reasoning in incorrect responses, while in others, it enhances both accuracy and reasoning efficiency. Understanding the underthinking phenomenon is crucial for developing models that not only provide correct answers but also exhibit effective reasoning processes.

## 3. Mitigating Underthinking Issues

In this section, we propose a lightweight mechanism that mitigates underthinking issues without requiring any model fine-tuning. Our experimental results using the QwQ-32B-Preview model demonstrate the effectiveness of this approach across all challenging test sets.

### 3.1. Decoding with Thought Switching Penalty

Aforementioned findings show that o1-like LLMs prioritize exploring many solutions over deeply investigating one. Inspired by the success of the coverage penalty in neural machine translation (Tu et al., 2016; Wu et al., 2016), we propose a novel decoding algorithm with a *thought switching penalty* to encourage the model to explore potential thoughts more thoroughly before moving on to new ones.

**Standard Decoding**  In standard decoding, the probability of each token $v$ at position $t$ is computed using the softmax function over the logits $\mathbf{z}_t \in \mathbb{R}^{|V|}$ (where $|V|$ is the vocabulary size) in the output layer:

$$P(x_t = v | x_{<t}) = \frac{\exp(z_{t,v})}{\sum_{v' \in V} \exp(z_{t,v'})}$$

where $z_{t,v} \in \mathbf{z}_t$ is the logit (unnormalized score) for token $v$. By repeating this step for each position in the sequence, the model generates sequences of tokens, computing probabilities for each possible continuation.

*Table 2.* Accuracy on AIME2022-2023 with respect to different values of $\alpha$ and $\beta$.

| Pass@1 Accuracy | | $\alpha$ | | | |
|---|---|---|---|---|---|
| | | 3 | 5 | 10 | 20 |
| $\beta$ | 300 | 35.2 | 37.0 | 39.0 | 39.4 |
| | 400 | 39.3 | 37.1 | 37.1 | 38.4 |
| | 500 | 38.5 | 38.7 | 39.1 | 39.2 |
| | 600 | **39.8** | 39.4 | 38.0 | 38.0 |
| | 700 | 37.1 | 39.4 | 39.0 | 38.3 |

**Thought Switching Penalty (TIP)** To encourage the model to delve deeper into current thoughts before switching, we introduce a penalty on tokens that are associated with thought transitions. Let $\widehat{V} \subset V$ be the set of tokens associated with thought switching (e.g., "alternatively"). We modify the logits as follows:

$$\hat{z}_{t,v} = \begin{cases} z_{t,v} - \alpha, & \text{if } v \in \widehat{V} \text{ and } t < \Psi + \beta \\ z_{t,v}, & \text{otherwise} \end{cases}$$

where

- $\alpha \geq 0$ (*Penalty Strength*) is a parameter controlling the strength of the penalty applied to thought-switching tokens. A larger $\alpha$ results in a greater reduction of the logits for these tokens, making them less likely to be chosen.

- $\beta \geq 0$ (*Penalty Duration*) specifies the number of positions from the start of a thought at $\Psi$, during which the penalty is active. A larger $\beta$ extends the penalty over more positions, further discouraging early thought switching.

When $\alpha = 0$ or $\beta = 0$, the penalty is effectively disabled, and the decoding process reduces to the standard decoding algorithm. The adjusted logits $\hat{z}_{t,v}$ reduce the probability of generating thought-switching tokens within a specified window, encouraging the model to continue expanding on the current thought before moving on.

The new probability distribution becomes:

$$\hat{P}(x_t = v \mid x_{<t}) = \frac{\exp(\hat{z}_{t,v})}{\sum_{v' \in V} \exp(\hat{z}_{t,v'})}$$

### 3.2. Experimental Results

We conducted the experiments using QwQ-32B-Preview, as the DeepSeek-R1-671B API does not allow for the modification of logits. To ensure a robust conclusion, we report Pass@1 results with four samples.

By adjusting $\alpha$ and $\beta$, we can control the model's behavior to achieve the desired level of thought exploration. We performed a grid search with $\alpha$ values in $[3, 5, 10, 20]$ and

*Table 3.* Results of the proposed decoding with TIP.

| Models | Pass@1 | |
|---|---|---|
| | Accuracy($\uparrow$) | UT Score ($\downarrow$) |
| *MATH500-Hard (Level 5)* | | |
| QwQ-32B-Preview | 82.8 | 71.1 |
| + TIP | 84.3 | 69.7 |
| *GPQA Diamond* | | |
| QwQ-32B-Preview | 57.1 | 59.1 |
| + TIP | 59.3 | 56.5 |
| *AIME2024* | | |
| QwQ-32B-Preview | 41.7 | 72.4 |
| + TIP | 45.8 | 68.2 |

$\beta$ values in $[300, 400, 500, 600, 700]$ using a development set that included the AIME 2022 and 2023 test sets. Table 2 lists the impact of varying the penalty strength $\alpha$ and penalty duration $\beta$ on the model's accuracy. We observe that increasing the penalty strength $\alpha$ generally leads to an improvement in accuracy up to a certain threshold, after which the benefits plateau or even diminish. Adjusting the penalty duration $\beta$ also significantly affects performance: At a lower penalty strength ($\alpha = 3$), increasing $\beta$ from 300 to 600 results in accuracy gains from 35.2% to 39.8%, the highest observed accuracy in our experiment. Conversely, at higher penalty strengths ($\alpha = 20$), extending $\beta$ beyond 300 leads to a decrease in accuracy, indicating that too long a penalty duration can hinder performance when combined with a strong penalty. We selected $\alpha = 3$ and $\beta = 600$ for our subquent experiments.

Table 3 lists the results of our approach in the three challenging test sets. Clearly, our approach consistently improves accuracy over the vanilla QwQ-32B-Preview in all cases by mitigating the underthinking issues. These consistent improvements across diverse and challenging datasets validate the effectiveness of the TIP approach in mitigating the underthinking issue identified in o1-like LLMs. By penalizing thought switches during decoding, TIP encourages the model to elaborate more thoroughly on each reasoning thought before considering alternative ones. This mechanism aligns with the human problem-solving process, where a focused and in-depth exploration of a particular approach often leads to correct solutions, especially in complex mathematical problem-solving contexts.

## 4. Related Work

### 4.1. Scaling Test-Time Compute

The advent of deep reasoning models, epitomized by OpenAI's o1, has sparked significant interest in scaling test-time

compute to enhance models' abilities to solve complex problems. Scaling test-time compute often involves two major strategies. The first is **expanding the search space**, which aims to broaden the scope of candidate solutions explored during decoding to ensure better final outcomes. Techniques in this category include self-consistency (Wang et al., 2023), where multiple answers are generated with a majority voting mechanism to select the final answer. Other methods include best-of-n decoding and minimum Bayes risk decoding (Lightman et al., 2024; Li et al., 2023; Khanov et al., 2024; Heineman et al., 2024; Wu et al., 2024).

The second direction, and arguably more transformative, focuses on **human-like deep thinking**. Efforts such as QwQ (Qwen, 2024), DeepSeek-R1 (DeepSeek, 2025) and Kimi-1.5 (Kimi, 2025), which aim to replicate OpenAI's o1, leverage reinforcement learning (RL) to endow models with advanced reasoning capabilities. Under large-scale RL training, these models exhibit emergent human-like thinking abilities characterized by deep, extended, and strategic reasoning. This allows them to explore diverse strategies, reflect on their decisions, revisit previous steps, and verify their conclusions. Such human-like thinking markedly improves accuracy, especially on complex reasoning tasks.

**Efficient Thinking**  Given that o1-like models aim to mimic human thought processes, the efficiency of their reasoning is critical to their performance on challenging problems. Just as human thinking can occasionally be inefficient, models may face similar issues. For instance, Chen et al. (2024) studied the problem of **overthinking** in o1-like LLMs, where models waste substantial computational resources revisiting trivial or self-evident paths, leading to inefficiency. Conversely, our focus lies on the underexplored problem of **underthinking**. Underthinking occurs when a model fails to deeply explore promising paths, instead frequently switching strategies prematurely, resulting in computational waste. This inefficiency becomes especially pronounced when tackling difficult problems. This phenomenon contrasts with overthinking, where excessive computational effort is invested in *simple problems* with diminishing returns. Underthinking refers to the model's tendency to abandon promising lines of reasoning prematurely on *challenging problems*, leading to incorrect answers. We assert that truly intelligent systems must learn to adaptively allocate their computational resources, concentrating on paths that are both promising and challenging.

### 4.2. Manipulating Decoding Penalties

The role of penalty mechanisms in Natural Language Processing (NLP) decoding has garnered significant attention. Traditional decoding methods, such as greedy search and beam search, focus primarily on maximizing the likelihood of generated sequences without considering the broader implications of the outputs. However, researchers have identified various shortcomings in these approaches, leading to the exploration of penalty mechanisms to enhance the quality of generated text.

Length normalization is a widely used strategy to adjust decoding penalties. Jean et al. (2015); Koehn & Knowles (2017); Tu et al. (2017); Murray & Chiang (2018) highlighted that length normalization and length penalties can prevent models from generating overly verbose or excessively brief translations, leading to improved fluency and adequacy. In addition, Tu et al. (2016) introduced coverage penalties in neural machine translation to mitigate the problems of "over-translation" and "under-translation" by integrating a coverage metric that penalizes repeated attention to tokens. Along this direction, Wu et al. (2016) proposed a coverage penalty in decoding to encourage the generation of an output that is most likely to cover all the words in the source sentence. See et al. (2017) incorporated the concept of coverage into the summarization task by modeling the coverage content in summarization outputs.

In this paper, we adjust decoding penalties to address the problem of underthinking. Our approach encourages the model to maintain its original line of reasoning and engage in deeper thought processes, avoiding frequent shifts in strategy and superficial reasoning patterns. To the best of our knowledge, we are the first to investigate the effectiveness of decoding penalties in mitigating the underthinking issue.

## 5. Conclusion

In this work, we investigated underthinking in o1-like LLMs, identifying it as a significant factor limiting their performance on challenging reasoning tasks. Through comprehensive analysis, we observed that these models frequently abandon promising reasoning paths prematurely, leading to inefficient problem-solving and lower accuracy. We introduced a novel metric to quantify underthinking by assessing token efficiency in incorrect responses. To mitigate this issue, we proposed a decoding strategy with thought switching penalty (TIP), which encourages models to thoroughly explore each reasoning thought before considering alternatives. Our empirical results demonstrate that TIP effectively reduces underthinking and enhances accuracy across difficult mathematical and scientific problem sets without necessitating additional model training.

This work contributes to a deeper understanding of reasoning processes in o1-like LLMs and provides a practical approach to align their problem-solving capabilities to human-like deep thinking. Future directions include exploring adaptive mechanisms within models to self-regulate thought transitions and further improving reasoning efficiency in o1-like LLMs.

## Impact Statement

The paper reveals the underthinking issue in o1-like models. Our findings call for the community to research on models' efficient thinking capabilities, which could significantly influence future developments in this field. We see no harmful impacts of this work.

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
