# OpenReview forum: "Thoughts Are All Over the Place: On the Underthinking of o1-Like LLMs"
_ICML.cc/2025/Conference — Submitted to ICML 2025_

### Official Review · Reviewer_4nho · 2025-03-09

**Overall Recommendation:** 3

**Summary:**

The paper propose an interesting phenomenon termed "underthinking" for LLM reasoning -- models frequently switch between different reasoning strategies without sufficiently exploring promising paths, leading to inadequate reasoning depth. Authors introduce a novel metric to quantify underthinking by measuring token efficiency in incorrect answers and propose a decoding strategy with a thought switching penalty to encourage deeper exploration of each reasoning path. Experimental results show that this strategy improves accuracy across challenging datasets without requiring model fine-tuning.

**Claims And Evidence:**

Yes, most claims supported by experiments.

**Essential References Not Discussed:**

There are no essential references missing from the paper.

**Experimental Designs Or Analyses:**

Yes. I believe the reliability of the authors' utilization of the Llama-3.3-70B model to automatically segment a response into reasoning thoughts needs further justification. Since many reasoning thoughts originate from more powerful reasoning models and tend to be very lengthy, the validity and quality of Llama's segmentation of reasoning thoughts should be discussed.

**Methods And Evaluation Criteria:**

Not quite reliable. A key step in evaluating underthinking is identifying whether early thoughts lead to the correct answer. However, the method used here leverages LLMs to assess whether each thought leads to a correct answer using a prompt. This approach is highly dependent on the capabilities of the chosen model and is significantly influenced by the prompt itself, potentially introducing bias. To ensure the objectivity of evaluation and analysis, the reliability of using LLMs to assess whether each thought leads to a correct answer needs further validation.

**Other Comments Or Suggestions:**

None.

**Other Strengths And Weaknesses:**

Strength
Good analysis on the phenomenon underthinking of LLMs, especially reasoning models.

Weaknesses:
The proposed Thought Switching Penalty method is heuristic and somewhat arbitrary. To encourage the model to explore its current thoughts more thoroughly before switching, the authors introduce a penalty on tokens associated with thought transitions, such as "alternatively." However, the impact of specific words varies across different models, which may make the effectiveness of this approach difficult to ensure. Moreover, there is insufficient exploration and analysis of how different words influence the results.

**Questions For Authors:**

What's the set of tokens for TIP? What's the influence of TIP on  inference time cost, i.e.,response length?

**Relation To Broader Scientific Literature:**

The key contributions of the paper are closely related to the broader scientific literature on large language models and their reasoning capabilities. It addresses the underexplored issue of underthinking.

**Theoretical Claims:**

No theoretical part in this paper.

---

> ### Author Rebuttal · Authors · 2025-04-01
>
> > Q1: Reliability of using LLMs to assess the correctness of intermediate reasoning steps.
>
> A1: We fully understand your concerns about the reliability of using LLMs to assess the correctness of intermediate reasoning steps, particularly the dependency on model capabilities and prompt sensitivity. In fact, we have explicitly addressed this issue in lines 246-258 of our submitted manuscript. As reported: "Our assessment approach achieves accuracies of 82.9% for correct examples and 81.8% for incorrect examples, demonstrating its effectiveness." These results offer quantitative validation of our evaluation method, although we agree further validation and model-independent analysis are valuable future work directions.
>
> ---
>
> > Q2: Reliability of using Llama-3.3-70B to automatically segment a response into reasoning thoughts.
>
> A2: Thank you very much for your insightful suggestion. We selected Llama-3.3-70B because, at the time of our ICML submission deadline, it was one of the strongest and most widely adopted open-source models available. To further validate our segmentation approach, we followed your recommendation and performed a manual evaluation on 100 randomly selected thought segments generated by DeepSeek-R1 on the AIME 2024 dataset. This manual check yielded an accuracy of 86%, further supporting the validity and effectiveness of our automated thought segmentation approach. We will explicitly report this additional validation result in the revised manuscript. Once again, we appreciate your valuable feedback.
>
> ---
>
> > Q3: Effectiveness of the proposed TIP method.
>
> A3: We have carefully addressed your points by conducting additional experiments to demonstrate the effectivenss and universality of the proposed TIP method. Below, we summarize our improvements with respect to your main comments:
>
> To further clarify the effectiveness of TIP, we conducted analyses on key reasoning metrics, specifically examining:
> - the average number of thought-switching tokens;
> - the length of intervals between thought switches.
>
> To examine TIP's generality explicitly, we reused hyperparameters tuned initially on the QwQ-32B-Preview model directly for the subsequent models — R1-Distill-Qwen-32B and DeepSeek-R1— without additional tuning.
>
> Our analysis across multiple benchmarks consistently showed that TIP significantly reduces the overall number of thought-switching tokens while increasing the average intervals between them. This suggests the models become more committed to exploring individual reasoning threads thoroughly before pivoting to alternatives, aligning with our design intent of addressing underthinking explicitly.
>
> In addition, TIP consistently and effectively enhanced reasoning performance across different models (R1-Distill-Qwen-32B, DeepSeek-R1) even without model-specific hyperparameter tuning. These results demonstrate both the universality and the empirical robustness of TIP.
>
> | Model|Pass@1|Pass@4|Pass@8|Pass@16|Thought Number|Thought Interval|Weighted UT Score (↓) |
> |-|-|-|-|-|-|-|-|
> |**MATH500-Hard(Lv5)**||||||||
> |QwQ-32B-Preview|83.1|92.4|94.4|95.8|12.6|445.6|11.7±20.5|
> |+TIP|83.7|93.2|95.3|96.4|5.7|517.6|11.0±19.5|
> |**GPQA Diamond**||||||||
> |QwQ-32B-Preview|57.6|78.5|85.3|90.3|21.1|356.8|25.1±23.9|
> |+TIP|59.1|78.9|85.8|91.2|7.3|432.5|23.2±23.2|
> |**AIME2024**||||||||
> |QwQ-32B-Preview|38.3|53.7|58.5|62.7|16.1|459.7|40.6±28.4|
> |+TIP|44.1|61.6|68.3|74.0|13.9|515.7|35.8±27.8|
> |R1-Distill-Qwen-32B|61.4|75.9|79.1|81.7|8.2|819.5|19.6±20.6|
> |+TIP|64.1|79.0|81.7|83.0|4.5|1018.0|17.7±20.6|
> |DeepSeek-R1|73.8|86.2|88.8|89.8|13.8|580.1|14.6±19.1|
> |+TIP|74.8|86.4|88.8|89.8|5.7|941.6|13.0±18.0|
>
> To thoroughly assess the robustness and generality of our TIP approach, we combined it with best-of-N sampling methods, and the results confirm that our method remains consistently effective under these conditions.
>
> |Models|Acc.(↑)|UT(↓)|
> |---|:---:|:---:|
> |QwQ+Self-Consistency|43.7|35.4|
> |+**TIP**|51.4|26.6|
> |R1-Distill-Qwen+Self-Consistency|67.0|13.4|
> |+**TIP**|69.9|12.5|
> |R1+Self-Consistency|79.3|10.1|
> |+**TIP**|81.3|7.5|

---

> > ### Comment · Reviewer_4nho · 2025-04-04
> >
> > Thanks for addressing my concerns, I have updated my score.

---

### Official Review · Reviewer_kam8 · 2025-03-11

**Overall Recommendation:** 3

**Summary:**

This paper introduces a novel investigation into the phenomenon of 'underthinking' in large language models (LLMs), specifically those designed for complex reasoning tasks, such as the 'o1-like' models. The authors define underthinking as the premature abandonment of promising reasoning paths, leading to suboptimal solutions. This is a significant contribution as it highlights a limitation in current models that has not been extensively studied. The core of their methodology involves identifying and segmenting 'thoughts' within the model's response, which they define as distinct reasoning steps, and then evaluating the correctness of each thought. To quantify underthinking, they propose a novel metric that measures token efficiency in incorrect responses, essentially assessing how many tokens are generated before a correct line of reasoning is abandoned. This metric provides a quantitative way to assess the efficiency of reasoning processes in LLMs, complementing traditional accuracy metrics. Furthermore, the authors introduce a decoding strategy called Thought Switching Penalty (TIP) to mitigate underthinking. TIP penalizes the model for switching between thoughts, encouraging it to explore each reasoning path more thoroughly before considering alternatives.

---

## After Rebuttal

**Score 2.0 -> 2.5**

**Claims And Evidence:**

The paper must provide more direct evidence of changes in the quality of reasoning, beyond just a reduction in thought switching. While the paper demonstrates improved accuracy, it's crucial to analyze whether this improvement stems from a genuine increase in the depth and detail of individual thoughts, as theoretically proposed, or merely from a decrease in the frequency of less productive thought shifts. Further analysis is needed to directly link the observed performance gains to the intended theoretical mechanism of promoting deeper, more sustained reasoning.

**Essential References Not Discussed:**

None.

**Experimental Designs Or Analyses:**

In my opinion, the supporting evidence from the experiments in this paper remains insufficient. The paper primarily focuses on the performance of the QwQ-32B-Preview model and three math and science tasks.

While the paper demonstrates improved accuracy and reduced thought switching, it is imperative to ascertain whether the remaining thoughts are genuinely deeper and more detailed, or if the improvement is solely a result of reduced switching frequency. Further analysis, including metrics on thought length, complexity, and reasoning steps per thought, is necessary to confidently claim that TSP promotes deeper thinking and not just less frequent thought switching.

**Methods And Evaluation Criteria:**

To enhance the practical applicability and ease of use of the TSP method, the current reliance on manual hyperparameter tuning needs to be addressed. While the paper provides a grid search example, a more systematic and robust approach to hyperparameter selection is required.

**Other Comments Or Suggestions:**

Honestly, for this work to truly reach its potential and make the broad impact I believe it can, you absolutely must consider expanding the experimental scope beyond just math and science problems. While the current findings are valuable, limiting ourselves here means we are potentially missing out on understanding if "underthinking" and TSP are relevant – and helpful – in other critical areas. Think about logical reasoning, everyday common sense, even creative tasks! Exploring these diverse areas isn't just about ticking boxes; it's about showing the real power and universal value of your insights to the whole research community.

From a truly responsible research perspective, we have to be cautious and explore the boundaries of TSP. While it's brilliant for addressing underthinking, I sincerely worry about scenarios where frequent thought switching might actually be a good thing – even necessary! We need to be able to honestly say when TSP is the right tool and when it might actually get in the way. Designing experiments to test this isn't just about being critical; it’s about being thorough and providing really useful guidance to anyone who wants to use TSP.

**Other Strengths And Weaknesses:**

This paper offers several noteworthy contributions to the field of large language models, particularly concerning complex reasoning. Firstly, the introduction of "underthinking" is a concept that I found to be quite insightful and genuinely novel. By pinpointing this tendency of LLMs to prematurely abandon potentially fruitful reasoning paths, the authors, I think, effectively highlight a key limitation in the current generation of these models. This isn't just about efficiency; it clearly impacts, and perhaps significantly, the accuracy of these models on really challenging problem-solving tasks. The formal definition and characterization of underthinking, especially with the empirical backing they provide, strikes me as a meaningful step towards a better understanding of how these models actually work internally.

The development of a quantitative metric for assessing underthinking is in my opinion, a valuable tool for researchers in this area. This proposed metric, measuring token efficiency in incorrect responses, provides a concrete way to delve deeper into a model's reasoning process beyond just simple accuracy scores. I think it allows for a more nuanced evaluation of model performance. The authors effectively demonstrate the utility of this metric by showing its correlation with the effectiveness of their proposed TSP method, which is a convincing demonstration.

**Questions For Authors:**

How does the proposed underthinking metric perform on tasks that require creative or divergent thinking, where the notion of a single "correct" answer may not be well-defined?

**Relation To Broader Scientific Literature:**

Good!

**Theoretical Claims:**

None.

---

> ### Author Rebuttal · Authors · 2025-04-01
>
> > Q1: More experimental results & analyses.
>
> A1: Thank you for your insightful feedback and constructive suggestions. We have carefully addressed your points in the revised manuscript and conducted additional experiments to substantiate our claims. Below, we summarize our improvements with respect to your main comments:
>
> To further clarify the effectiveness of TIP, we conducted analyses on key reasoning metrics, specifically examining:
> - the average number of thought-switching tokens;
> - the length of intervals between thought switches.
>
> To examine TIP's generality explicitly, we reused hyperparameters tuned initially on the QwQ-32B-Preview model directly for the subsequent models — R1-Distill-Qwen-32B and DeepSeek-R1— without additional tuning.
>
> Our analysis across multiple benchmarks consistently showed that TIP significantly reduces the overall number of thought-switching tokens while increasing the average intervals between them. This suggests the models become more committed to exploring individual reasoning threads thoroughly before pivoting to alternatives, aligning with our design intent of addressing underthinking explicitly.
>
> In addition, TIP consistently and effectively enhanced reasoning performance across different models (R1-Distill-Qwen-32B, DeepSeek-R1) even without model-specific hyperparameter tuning. These results demonstrate both the universality and the empirical robustness of TIP.
>
> | Model|Pass@1|Pass@4|Pass@8|Pass@16|Thought Number|Thought Interval|Weighted UT Score (↓) |
> |-|-|-|-|-|-|-|-|
> |**MATH500-Hard(Lv5)**||||||||
> |QwQ-32B-Preview|83.1|92.4|94.4|95.8|12.6|445.6|11.7±20.5|
> |+TIP|83.7|93.2|95.3|96.4|5.7|517.6|11.0±19.5|
> |**GPQA Diamond**||||||||
> |QwQ-32B-Preview|57.6|78.5|85.3|90.3|21.1|356.8|25.1±23.9|
> |+TIP|59.1|78.9|85.8|91.2|7.3|432.5|23.2±23.2|
> |**AIME2024**||||||||
> |QwQ-32B-Preview|38.3|53.7|58.5|62.7|16.1|459.7|40.6±28.4|
> |+TIP|44.1|61.6|68.3|74.0|13.9|515.7|35.8±27.8|
> |R1-Distill-Qwen-32B|61.4|75.9|79.1|81.7|8.2|819.5|19.6±20.6|
> |+TIP|64.1|79.0|81.7|83.0|4.5|1018.0|17.7±20.6|
> |DeepSeek-R1|73.8|86.2|88.8|89.8|13.8|580.1|14.6±19.1|
> |+TIP|74.8|86.4|88.8|89.8|5.7|941.6|13.0±18.0|
>
> To thoroughly assess the robustness and generality of our TIP approach, we combined it with best-of-N sampling methods, and the results confirm that our method remains consistently effective under these conditions.
>
> |Models|Acc.(↑)|UT(↓)|
> |---|:---:|:---:|
> |QwQ+Self-Consistency|43.7|35.4|
> |+**TIP**|51.4|26.6|
> |R1-Distill-Qwen+Self-Consistency|67.0|13.4|
> |+**TIP**|69.9|12.5|
> |R1+Self-Consistency|79.3|10.1|
> |+**TIP**|81.3|7.5|
>
> ---
>
> > Q2: How does the proposed underthinking metric perform on tasks that require creative or divergent thinking, where the notion of a single "correct" answer may not be well-defined?
>
> A2: We fully acknowledge the importance and potential value of broadening our evaluations beyond purely math and science tasks. Your thoughtful comment highlights critical considerations regarding the broader relevance of the underthinking concept and the TIP method.
>
> In this work, closely following the technical approach in the DeepSeek-R1 study [1], we concentrated primarily on reasoning-intensive quantitative/formal knowledge tasks, as these tasks are directly aligned with clearly defined correctness criteria and enable systematic measurement and analysis of underthinking behaviors in a reproducible manner.
>
> We recognize that exploring more diverse reasoning tasks, such as logical reasoning and common-sense scenarios, could amplify our method's scope and impact. While tasks requiring creative or open-ended thinking represent valuable but challenging new directions (due to ambiguous evaluation criteria and limitations in large-scale reinforcement learning methods), we aim to pursue these avenues in follow-up studies. Carefully investigating these scenarios will indeed help establish clearer boundaries regarding when TIP is beneficial, when frequent thought switching might be advantageous, and under which conditions it might even be necessary.
>
> We will explicitly discuss this direction as an important avenue for future work in the revised manuscript. For now, the current scope is clearly declared and motivated as an initial, controlled investigation into the robustness and effectiveness of addressing underthinking in reproducibly measurable tasks.
>
>  [1] DeepSeek. "Deepseek-r1: Incentivizing reasoning capability in llms via reinforcement learning." arXiv preprint arXiv:2501.12948 (2025).

---

> > ### Comment · Reviewer_kam8 · 2025-04-03
> >
> > Thank you for your reply, which has solved my problem to some extent!
> >
> > If other reviewers are willing to accept this paper, then I will not object. In my mind, the rating of this paper is at the borderline, which is 2.5.

---

### Official Review · Reviewer_cw7i · 2025-03-14

**Overall Recommendation:** 3

**Summary:**

This paper investigates strategies to leverage decoding-time interventions for improving reasoning depth and accuracy in o1-like LLMs. The authors propose a thought switching penalty TIP decoding strategy that discourages frequent and premature switching between reasoning paths to mitigate underthinking; then, they evaluate TIP across multiple complex reasoning benchmarks using two o1-like LLMs. They find significant gains in task accuracy and reasoning depth compared to baseline greedy decoding and standard sampling methods.

**Claims And Evidence:**

The authors' claim that TIP improves reasoning depth and accuracy is generally supported by experimental results that show consistent performance gains over baseline decoding. However, the paper does not provide a deeper analysis of why TIP works across different types of reasoning tasks and whether it may negatively affect tasks requiring flexible thought switching.

**Essential References Not Discussed:**

I find the related work discussion comprehensive, and I do not see essential references missing that would significantly impact the contributions

**Experimental Designs Or Analyses:**

The experiments are extensive, with detailed analysis of the results. These experiments validate the effectiveness of TIP in improving reasoning quality and task accuracy and demonstrate the robustness and scalability of the method across different sizes and complexities of benchmarks.

**Methods And Evaluation Criteria:**

The workflow is well-structured, as it combines a novel underthinking metric with decoding adjustments to make the intervention measurable and actionable. This approach quantifies the problem of underthinking in a transparent way and further provides a practical solution that operates entirely at inference time, making it broadly applicable to existing o1-like LLMs.

**Other Comments Or Suggestions:**

I suggest the authors include a case study or a clearer explanation and analysis of Figure 2 to better illustrate how TIP affects reasoning depth and thought switching in practice.

**Other Strengths And Weaknesses:**

The paper proposes TIP as a decoding strategy to improve reasoning quality and shows its empirical effectiveness. However, the authors do not provide a theoretical discussion on how TIP relates to reasoning depth. My understanding is that TIP is designed based on the intuition that reducing thought switching encourages deeper reasoning. If the authors could offer a formal justification, quantitative analysis, or even case studies to support this design choice, it would significantly strengthen the credibility of the method.

**Questions For Authors:**

I wonder how TIP balances preventing shallow jumps with avoiding overly rigid reasoning. How is the generality of TIP across different tasks, especially those that require flexible or multi-path reasoning? Does it have potential downsides, such as limiting model adaptability?

**Relation To Broader Scientific Literature:**

The idea of improving LLM reasoning depth through decoding-time interventions to address underthinking is novel and timely. Most existing work focuses on model fine-tuning or prompting strategies to improve reasoning, but these approaches cannot control reasoning behavior at inference time, limiting their flexibility and applicability to fixed models. This paper introduces a method that applies a thought switching penalty during decoding to encourage deeper exploration within each line of thought, as well as significantly improves reasoning accuracy and coherence without requiring model retraining.

**Theoretical Claims:**

The paper does not present much formal theoretical proofs. It mainly relies on an empirical definition of underthinking and the proposed TIP mechanism. I find the explanation of underthinking and thought switching intuitive and reasonable, but I believe a formal analysis of how TIP quantitatively relates to reasoning depth would strengthen the theoretical grounding of the method.

---

> ### Author Rebuttal · Authors · 2025-04-01
>
> > Q1: More insights about how TIP works and the generality of TIP.
>
> A1: Thank you for your insightful questions and constructive feedback. We fully agree that more in-depth insights into *why* TIP enhances models' reasoning depth and accuracy would significantly reinforce our method.
>
> To further clarify the effectiveness of TIP, we conducted analyses on key reasoning metrics, specifically examining:
> - the average number of thought-switching tokens;
> - the length of intervals between thought switches.
>
> To examine TIP's generality explicitly, we reused hyperparameters tuned initially on the QwQ-32B-Preview model directly for the subsequent models — R1-Distill-Qwen-32B and DeepSeek-R1— without additional tuning.
>
> Our analysis across multiple benchmarks consistently showed that TIP significantly reduces the overall number of thought-switching tokens while increasing the average intervals between them. This suggests the models become more committed to exploring individual reasoning threads thoroughly before pivoting to alternatives, aligning with our design intent of addressing underthinking explicitly.
>
> In addition, TIP consistently and effectively enhanced reasoning performance across different models (R1-Distill-Qwen-32B, DeepSeek-R1) even without model-specific hyperparameter tuning. These results demonstrate both the universality and the empirical robustness of TIP.
>
> | Model|Pass@1|Pass@4|Pass@8|Pass@16|Thought Number|Thought Interval|Weighted UT Score (↓) |
> |-|-|-|-|-|-|-|-|
> |**MATH500-Hard(Lv5)**||||||||
> |QwQ-32B-Preview|83.1|92.4|94.4|95.8|12.6|445.6|11.7±20.5|
> |+TIP|83.7|93.2|95.3|96.4|5.7|517.6|11.0±19.5|
> |**GPQA Diamond**||||||||
> |QwQ-32B-Preview|57.6|78.5|85.3|90.3|21.1|356.8|25.1±23.9|
> |+TIP|59.1|78.9|85.8|91.2|7.3|432.5|23.2±23.2|
> |**AIME2024**||||||||
> |QwQ-32B-Preview|38.3|53.7|58.5|62.7|16.1|459.7|40.6±28.4|
> |+TIP|44.1|61.6|68.3|74.0|13.9|515.7|35.8±27.8|
> |R1-Distill-Qwen-32B|61.4|75.9|79.1|81.7|8.2|819.5|19.6±20.6|
> |+TIP|64.1|79.0|81.7|83.0|4.5|1018.0|17.7±20.6|
> |DeepSeek-R1|73.8|86.2|88.8|89.8|13.8|580.1|14.6±19.1|
> |+TIP|74.8|86.4|88.8|89.8|5.7|941.6|13.0±18.0|
>
> ---
>
> > Q2: I suggest the authors include a case study or a clearer explanation and analysis of Figure 2 to better illustrate how TIP affects reasoning depth and thought switching in practice.
>
> A2: Thank you for this valuable suggestion. Due to length constraints, we did not include the complete response within the paper nor clearly specified whether each thought was promising. In the revised manuscript, we will provide the full response in an appendix and enhance the clarity and interpretability of Figure 2 accordingly.

---

### Official Review · Reviewer_nvHv · 2025-03-14

**Overall Recommendation:** 1

**Summary:**

This paper introduces and investigates the phenomenon of "underthinking" in advanced "o1-like" large language models, characterized by their tendency to frequently switch between reasoning thoughts without sufficiently exploring promising paths, particularly on complex problems. The authors empirically demonstrate that this behavior correlates with incorrect responses, which often exhibit higher thought counts and token usage compared to correct ones.

# update after rebuttal

Since most of my concerns remain and significant revisions are still needed, I strongly suggest another round of revision. Thanks.

**Claims And Evidence:**

Fine

**Essential References Not Discussed:**

Fine

**Experimental Designs Or Analyses:**

Fine

**Methods And Evaluation Criteria:**

Fine

**Other Comments Or Suggestions:**

N/A

**Other Strengths And Weaknesses:**

Hope the feedback is valuable for the authors and helps improve the quality in the revision or camera ready. I am happy to update my score after rebuttal if necessary. Thanks!

Pros:
1. Clearly defines and names "underthinking" as a distinct reasoning inefficiency in advanced LLMs, contrasting it with potential overthinking.
2. Addresses a critical aspect of state-of-the-art reasoning models (o1-like capabilities) and their practical limitations.
3. Provides quantitative analysis (token/thought counts, correlations) across multiple challenging datasets (MATH, GPQA, AIME) and relevant models (Qwen, DeepSeek variants).
4. The Underthinking (UT) score offers a novel, quantitative way to measure a specific type of reasoning inefficiency in incorrect responses.

Cons:
1. It is suggested that the authors provide the code for reproduction.
2. The term "o1-like" relies heavily on community understanding and specific named replicas. While practical, briefly defining the key characteristics being emulated (e.g., explicit iterative thought generation, test-time compute scaling structure) beyond just "deep reasoning" would strengthen the paper's foundation and clarify the scope.
3. It is not very clear how the authors set the hyperparameters such as temperatures. It suggested that the authors provide more details about all the experiment setting.
4. It is not very clear about the impact of hyperparameters such as temperatures on the "Underthinking" issues. It is suggested that the authors conduct a more detailed analysis on the sensitivity of the hyperparameters.
5. There is no formal definition of the "Underthinking" issue. It is suggested the authors provide a formal definition of the "Overthinking" issue. Otherwise, it is not very clear. The current description "where o1-like LLMs frequently switch between different reasoning thoughts without sufficiently exploring promising paths to reach a correct solution." is kind of vague. What do "different reasoning thoughts" and "promising paths" refer to?
6. It is not very clear how often the "Underthinking" issue happens. What is the percentage of the "Underthinking" issue in all the wrong responses? A more quantitive analysis is desired.
7. Figure 2 needs more explanations. Is there any thought is "promising"? It the final answer correct or not?
8. The insight is limited and the overall observations are superficial. Why Some LLMs have the "Underthinking" issues? Do all reasoning LLMs have the "Underthinking" issues? What are the fundamental cause of the "Underthinking" issues? Does the proposed method solve the fundamental cause of the "Underthinking" issues?
9. It is suggested that the authors provide more example to how whether the proposed method really mitigate the "Underthinking" issue. Otherwise, it may be not convincing enough.

**Questions For Authors:**

See Weaknesses

**Relation To Broader Scientific Literature:**

Fine

**Theoretical Claims:**

Fine

---

> ### Author Rebuttal · Authors · 2025-04-01
>
> Thank you for the valuable suggestions, which helped clarify and strengthen the paper.
>
> > Q1: Provide the code for reproduction.
>
> A1: We will make our code publicly available soon.
>
> ---
>
> > Q2: The term "o1-like models" isn't precisely defined.
>
> A2: Thank you for pointing this out. We acknowledge that our original description of “o1-like” models lacks precision. A clearer term would be “Long Reasoning Models”, which refers to models that generate detailed CoT reasoning by iteratively producing intermediate reasoning steps and sequentially refining solutions until reaching a final conclusion.
>
> ---
>
> > Q3: More details about the temperatures.
>
> A3: In our experiments, we set the temperature to 0.7 and the top-p value to 0.95.
>
> We followed your suggestion and conducted a more detailed analysis regarding the impact of temperature settings. The table below summarizes the results of QwQ-32B-Preview evaluated on the Math Level 1 and 5. Clearly, the temperature hyperparameter has only a marginal effect on the number of generated thoughts, further confirming our assertion that "underthinking" is a fundamental behavior inherent to these models.
>
> |Level|Temperature|Thoughts|
> |---|---|---|
> |1|0|3.7|
> |1|0.3|4.6|
> |1|0.5|3.8|
> |1|0.7|4.1|
> |1|1|3.7|
> ||||
> |5|0|10.9|
> |5|0.3|11.1|
> |5|0.5|11.6|
> |5|0.7|10.8|
> |5|1|10.7|
>
> ---
>
> > Q4: There is no formal definition of the "Underthinking" issue.
>
> A4: We recognize the terminology requires clarity. We clarify definitions explicitly:
>
> - Underthinking (formal definition): Defined in Section 2.3 as: "Underthinking occurs when models generate potentially correct intermediate reasoning thoughts initially, but prematurely shift away without thoroughly exploring and continuing these promising reasoning paths, eventually yielding incorrect answers."
> - Different Reasoning Thoughts: Defined explicitly in Section 2.1 as: "Individual cognitive steps within a model’s reasoning strategy. These transitions often explicitly indicated through terms like "alternatively"."
> - Promising Paths: Defined in Section 2.2 as: Reasoning trajectories identified as promising based on intermediate correctness evaluations. We describe our detailed implementation and assessment strategies for thought correctness clearly in that section.
>
> ---
>
> > Q5: The percentage of underthinking issue in all the wrong responses.
>
> A5: We presented in Table 1 the quantitative results of the underthinking score, which focuses the token efficiency specifically on the wrong responses. The results show that all Long Reasoning Models suffer from significant underthinking issues.
>
> ---
>
>  > Q6: Figure 2 needs more explanations.
>
> A6: The final correct answer for the case in Figure 2 is 23. Thoughts #1,3,4,5 are promising and can lead to the correct answer if explored sufficiently.
>
> ---
>
> > Q7: The insight is limited and the overall observations are superficial.
>
> A7: In this pioneering work, our main contributions include:
> - Clearly defining and evaluating underthinking in long reasoning models.
> - Introducing quantitative evaluation measures.
> - Proposing a simple yet effective decoding approach that significantly mitigates underthinking on multiple reasoning benchmarks such as Math500, GPQA Diamond, and AIME2024.
>
> At this stage, we acknowledge that the existing community lacks complete open-source reproductions of large-scale long reasoning models, making it prohibitively expensive to empirically investigate training-related foundational causes. Therefore, we do not claim contributions toward understanding the training mechanism behind underthinking. However, clearly defining the issue and presenting empirical mitigation techniques is a valuable foundational step that future studies can further build upon toward fundamental analyses.
>
> ---
>
> > Q8: More convinced analyses.
>
> A8: We conducted extensive experiments across multiple models on the AIME24 benchmark. The results demonstrate that our proposed approach significantly reduces both the average number of thoughts generated and the average length of each thought, underscoring the broad effectiveness of our method.
>
> |Models|Pass@1|Number|Interval|
> |------|------|------|--------|
> |QwQ-32B-Preview|38.3|16.1|459.7|
> |+TIP|44.1|13.9|515.7|
> |R1-Distill-Qwen-32B|61.4|8.2|819.5|
> |+TIP|64.1|4.5|1018.0|
> |DeepSeek-R1|73.8|13.8|580.1|
> |+TIP|74.8|5.7|941.6|
>
> To thoroughly assess the robustness and generality of our TIP approach, we combined it with best-of-N sampling methods, and the results confirm that our method remains consistently effective under these conditions.
>
> |Models|Acc.(↑)|UT(↓)|
> |---|:---:|:---:|
> |QwQ+Self-Consistency|43.7|35.4|
> |+**TIP**|51.4|26.6|
> |R1-Distill-Qwen+Self-Consistency|67.0|13.4|
> |+**TIP**|69.9|12.5|
> |R1+Self-Consistency|79.3|10.1|
> |+**TIP**|81.3|7.5|

---

### Decision · Program_Chairs · 2025-05-01

**Decision:**

Reject

**Comment:**

The paper introduces a decoding strategy called TIP to address the "under thinking" issue in LLMs. The authors found that LLMs can switch between reasoning paths without deeply exploring a specific solution. The method penalizes thought transitions during decoding, encouraging the model to stick with and develop a single reasoning path more thoroughly. TIP has achieved strong performances and is training free. TIP is based on manually selected transition tokens. In the discussion phase, the reviewers mentioned that the submission could benefit from another round of revision to improve its clarity and substantiate its claims including more tasks and a wider selection of models.